# Interpretable factorization of clinical questionnaires to identify latent factors of psychopathology

## Abstract

Psychiatry research seeks to understand the manifestations of psychopathology in behavior, as measured in questionnaire data, by identifying a small number of latent factors that explain them. While factor analysis is the traditional tool for this purpose, the resulting factors may not be interpretable, and may also be subject to confounding variables. Moreover, missing data are common, and explicit imputation is often required. To overcome these limitations, we introduce interpretability constrained questionnaire factorization (ICQF), a non-negative matrix factorization method with regularization tailored for questionnaire data. Our method aims to promote factor interpretability and solution stability. We provide an optimization procedure with theoretical convergence guarantees, and an automated procedure to detect latent dimensionality accurately. We validate these procedures using realistic synthetic data. We demonstrate the effectiveness of our method in a widely used general-purpose questionnaire, in two independent datasets (the Healthy Brain Network and Adolescent Brain Cognitive Development studies). Specifically, we show that ICQF improves interpretability, as defined by domain experts, while preserving diagnostic information across a range of disorders, and outperforms competing methods for smaller dataset sizes. This suggests that the regularization in our method matches domain characteristics.

## 1 Introduction

Standardized questionnaires are a common tool in psychiatric practice and research, for purposes ranging from screening to diagnosis or quantification of severity. A typical questionnaire comprises questions – usually referred to as *items* – reflecting the degree to which particular symptoms or behavioural issues are present in study participants. Items are chosen as evidence for the presence of *latent constructs* giving rise to the psychiatric problems observed. For many common disorders, there is a practical consensus on constructs. If so, a questionnaire may be organized so that subsets of the items can be added up to yield a *subscale score* quantifying the presence of their respective construct. Otherwise, the goal may be to discover constructs through factor analysis.

The *factor analysis* (FA) of a questionnaire matrix (#participants $\times$ #items) expresses it as the product of a factor matrix (#participants $\times$ #factors) and a loading matrix (#factors $\times$ #items). The method assumes that answers to items should be correlated, and can therefore be explained in terms of a smaller number of factors. The method yields two real-valued matrices, with uncorrelated columns in the factor matrix. The number of factors is specified a priori, or estimated from data. The values of the factors for each participant can then be viewed as a succinct representation of them.

Interpreting what construct a factor may represent is done by considering its loadings across items. Ideally, if very few items have a non-zero loading, or each item only has a high loading on a single factor, it will be easy to associate the factor with them. The FA solution is often subjected to rotation

to try to accomplish this. In practice, the loadings could be an arbitrary linear combination of items, with positive and negative weights. Factors are real-valued, and neither their magnitude nor their sign are intrinsically meaningful. Beyond this, any missing data will have to be imputed, or the respective items omitted, before FA can be used. Finally, patterns in answers that are driven by other characteristics of participants (e.g. age or sex) are absorbed into factors themselves, acting as confounders, instead of being represented separately or controlled for.

In this paper, we propose to address all of the issues above with a novel matrix factorization method specifically designed for use with questionnaire data, through the following contributions:

**1. Interpretability-Constrained Questionnaire Factorization (ICQF)** Our method incorporates key characteristics which enhance the interpretability of resulting factors, as conveyed by clinical psychiatry collaborators. These characteristics are translated into mathematical constraints as follows:

- Factor values are within the range of $[0, 1]$, representing the degree of presence of the factor.

- Factor loadings are bounded within the same range as the original questionnaire responses, facilitating interpretation as answer patterns associated with the factor, rather than arbitrary values.

- The reconstructed matrix adheres to the range or observed maximum of the original questionnaire, preventing any entry from exceeding these limits.

- The method directly handles missing data without requiring imputation. Additionally, it allows for the inclusion of pre-specified factors to capture answer patterns correlated with known variables.

**2. Theoretical foundations of ICQF** Introducing constraints on both the factors and the reconstructed matrix poses algorithmic challenges. We introduce an optimization procedure for ICQF, using alternating minimization with ADMM, and we demonstrate that it converges to a local minimum of the optimization problem. We implement blockwise-cross-validation (BCV) to determine the number of factors. We show that, if this number of factors is close to that underlying the data, the solution will be close to a global minimum. We also empirically demonstrate that BCV detects the number of factors more precisely than competing methods through synthetic questionnaire examples.

**3. Method evaluation** We conduct a comprehensive evaluation of ICQF in comparison with state-of-the-art methods on CBCL, a widely used questionnaire to assess behavioral and emotional problems, collected in two independent clinical studies (*HBN* and *ABCD*). We demonstrate the effectiveness of our method on quantitative metrics that reflect preservation of diagnostic information in latent factors, and stability of factor loadings in limited sample sizes or across datasets.

**4. Light-weighted implementation** We provide a Python implementation of ICQF that can efficiently handle typical questionnaire datasets in psychology or psychiatry research contexts.

## 2   Related Work and Technical Motivation for our Method

The extraction of latent variables (a.k.a. factors) from matrix data is often done through low rank matrix factorizations, such as singular value decomposition (SVD), principal component analysis (PCA) and exploratory Factor Analysis (hereafter, just FA) (Golub & Van Loan, 2013; Bishop & Nasrabadi, 2006). While SVD and PCA aim at reconstructing the data, FA aims at explaining correlations between (questions) items through latent factors (Bandalos & Boehm-Kaufman, 2010). Factor rotation (Browne, 2001; Sass & Schmitt, 2010; Schmitt & Sass, 2011) is then performed to obtain a sparser solution which is easier to interpret and analyze. For a review of FA, see Thompson (2004); Gaskin & Happell (2014); Gorsuch (2014); Goretzko et al. (2021). Non-negative matrix factorization (NMF) was proposed as a way of identifying sparser, more interpretable latent variables, which can be added to reconstruct the data matrix. It was introduced in Paatero & Tapper (1994) and developed in Lee & Seung (2000). Different varieties of NMF-based models have been proposed for various applications, such as the sparsity-controlled (Eggert & Korner, 2004; Qian et al., 2011), manifold-regularized (Lu et al., 2012), orthogonal Ding et al. (2006); Choi (2008), convex/semi-convex (Ding et al., 2008), or archetypal regularized NMF (Javadi & Montanari, 2020). More recently, Deep-NMF (Trigeorgis et al., 2016; Zhao et al., 2017) and Deep-MF (Xue et al., 2017; Fan & Cheng, 2018; Arora et al., 2019) can model non-linearities on top of (non-negative) factors, when the sample is large (Fan, 2021). These methods do not directly model either the interpretability characteristics or the constraints that we view as desirable. If the goal is to identify latent variables relevant for multiple matrices, the standard approach is multi-view learning (Sun et al., 2019), or variants that

89 can handle only partial overlap in participants across matrices (Ding et al., 2014; Gunasekar et al.,
90 2015; Gaynanova & Li, 2019). Finally, non-negative matrix tri-factorization (Li et al., 2009; Pei et al.,
91 2015), supports an additional matrix mapping between latent representations for different matrices.

92 Obtaining a factorization with these methods requires both specifying the number of latent variables,
93 and solving an optimization problem. In SVD/PCA, the number of variables is often selected based
94 on the percentage of variance explained, or determined via techniques such as spectral analysis, the
95 Laplace-PCA method, or Velicer's MAP test (Velicer, 1976; Velicer et al., 2000; Minka, 2000). For
96 FA, several methods have been proposed: Bartlett's test (Bartlett, 1950), parallel analysis (Horn, 1965;
97 Hayton et al., 2004), MAP test and comparison data (Ruscio & Roche, 2012). For NMF, iterative
98 detection algorithms are recommended, e.g. the Bayesian information criterion (BIC) (Stoica &
99 Selen, 2004), cophenetic correlation coefficient (CCC) (Fogel et al., 2007) and the dispersion (Brunet
100 et al., 2004). More recent proposals for NMF are Bi-cross-validation (BiCV) (Owen & Perry, 2009)
101 and its generalization, the blockwise-cross-validation (BCV) (Kanagal & Sindhwani, 2010), which
102 we use in this paper. The optimization problem for NMF is non-convex, and different algorithms for
103 solving it have been proposed. Multiplicative update (MU) (Lee & Seung, 2000) is the simplest and
104 mostly used. Projected gradient algorithms such as the block coordinate descent (Cichocki & Phan,
105 2009; Xu & Yin, 2013; Kim et al., 2014) and the alternating optimization (Kim & Park, 2008; Mairal
106 et al., 2010) aim at scalability and efficiency in larger matrices. Given that our optimization problem
107 has various constraints, we use a combination of alternative optimization and Alternating Direction
108 Method of Multipliers (ADMM) (Boyd et al., 2011; Huang et al., 2016).

## 3 Methods

### 3.1 Interpretable Constrained Questionnaire Factorization (ICQF)

111 **Inputs** Our method operates on a questionnaire data matrix $M \in \mathbb{R}_{\geq 0}^{n \times m}$ with $n$ participants and
112 $m$ questions, where entry $(i, j)$ is the answer given by participant $i$ to question $j$. As questionnaires
113 often have missing data, we also have a mask matrix $\mathcal{M} \in \{0, 1\}^{n \times m}$ of the same dimensionality
114 as $M$, indicating whether each entry is available $(= 1)$ or not $(= 0)$. Optionally, we may have a
115 confounder matrix $C \in \mathbb{R}_{\geq 0}^{n \times c}$, encoding $c$ known variables for each participant that could account for
116 correlations across questions (e.g. age or sex). If the $j^{th}$ confound $C_{[:,j]}$ is categorical, we convert
117 it to indicator columns for each value. If it is continuous, we first rescale it into $[0, 1]$ (where 0 and
118 1 are the minimum and maximum in the dataset), and replace it with two new columns, $C_{[:,j]}$ and
119 $1 - C_{[:,j]}$. This mirroring procedure ensures that both directions of the confounding variables are
120 considered (e.g. answer patterns more common the younger or the older the participants are). Lastly,
121 we incorporate a vector of ones into $C$ to facilitate intercept modeling of dataset wide answer patterns.

122 **Optimization problem** We seek to factorize the questionnaire matrix $M$ as the product of a
123 $n \times k$ factor matrix $W \in [0, 1]$, with the confound matrix $C \in [0, 1]$ as optional additional columns,
124 and a $m \times (k + c)$ loading matrix $Q := [^R Q, ^C Q]$, with a loading pattern $^R Q$ over $m$ questions for
125 each of the $k$ factors (and $^C Q$ for optional confounds). Denoting the Hadamard product as $\odot$, our
126 optimization problem minimizes the squared error of this factorization

$$
\begin{aligned}
\underset{W \in \mathcal{W}, Q \in \mathcal{Q}, Z \in \mathcal{Z}}{\text{minimize}} \quad & 1/2 \left\| \mathcal{M} \odot (M - Z) \right\|_F^2 + \beta \cdot R(W, Q) \\
\text{such that} \quad & [W, C] Q^T = Z, \ \mathcal{Z} = \{Z | \min(M) \leq Z_{ij} \leq \max(M)\} \\
& \mathcal{Q} = \{Q | 0 \leq Q_{ij}\} \text{ and } \mathcal{W} = \{W | 0 \leq W_{ij} \leq 1\}
\end{aligned}
\tag{ICQF}
$$

127 subject to entries of $Q$ being in the same value range as question answers, so loadings are interpretable,
128 and bounding the reconstruction by the range of values in the questionnaire matrix $M$. We further
129 regularize $W$ and $Q$ through $R(W, Q) := \|W\|_{p,q} + \gamma \|Q\|_{p,q}$, $\gamma = \frac{n}{m} \max(M)$, where $\|A\|_{p,q} :=$
130 $(\sum_{i=1}^m (\sum_{j=1}^n |A_{ij}|^p)^{q/p})^{1/q}$. Here, we use $p = q = 1$ for sparsity control. The heuristic $\gamma$ balances
131 the sparsity control between $W$ and $Q$; $\gamma$ is absorbed into $\beta$ of $Q$ if no ambiguity results.

### 3.2 Solving the optimization problem

133 We use the ADMM framework for fitting the ICQF model, due to its parallelizability, flexibility in
134 incorporating various types of constraints, and its compatibility with different optimization schemes.

Specifically, we utilize the Fast Iterative Shrinkage Thresholding Algorithm (FISTA) to accommodate our sparsity constraints, leveraging its numerical advantages, such as quadratic convergence and low memory cost, as discussed in Gaines et al. (2018). Unlike stochastic optimization approaches, which require addressing the missing entries and uneven distribution of responses in questionnaires when generating training batches, ADMM allows us to tackle the optimization problem holistically. Additionally, it can find a solution for large clinical questionnaire datasets (thousands of participants, tens to hundreds of questions) in about a minute with a laptop CPU, so the performance is appropriate.

**Optimization procedure**   The ICQF problem is non-convex and requires satisfying multiple constraints. Under the ADMM optimization procedure, the Lagrangian $\mathcal{L}_\rho$ is:

$$
\begin{aligned}
\mathcal{L}_\rho(W, Q, Z, \alpha_Z) =& 1/2\|\mathcal{M} \odot (M - Z)\|_F^2 + \mathcal{I}_\mathcal{W}(W) + \beta\|W\|_{1,1} + \mathcal{I}_\mathcal{Q}(Q) + \beta\|Q\|_{1,1} \\
& + \langle \alpha_Z, Z - [W, C]Q^T \rangle + \rho/2 \left\| Z - [W, C]Q^T \right\|_F^2 + \mathcal{I}_\mathcal{Z}(Z)
\end{aligned}
\tag{1}
$$

where $\rho$ is the penalty parameter, $\alpha_Z$ is the vector of Lagrangian multipliers and $\mathcal{I}_\mathcal{X}(X) = 0$ if $X \in \mathcal{X}$ and $\infty$ otherwise. We alternately update primal variables $W, Q$ and the auxiliary variable $Z$ by solving the following sub-problems:

$$
W^{(i+1)} = \underset{W \in \mathcal{W}}{\arg\min} \; \rho/2\|Z^{(i)} - [W, C]Q^{(i),T} + \rho^{-1}\alpha_Z^{(i)}\|_F^2 + \beta\|W\|_{1,1}
\tag{2}
$$

$$
Q^{(i+1)} = \underset{Q \in \mathcal{Q}}{\arg\min} \; \rho/2\|Z^{(i)} - [W^{(i+1)}, C]Q^T + \rho^{-1}\alpha_Z^{(i)}\|_F^2 + \beta\|Q\|_{1,1}
\tag{3}
$$

$$
Z^{(i+1)} = \underset{Z \in \mathcal{Z}}{\arg\min} \; \|\mathcal{M} \odot (M - Z)\|_F^2 + \rho\|Z - [W^{(i+1)}, C]Q^{(i+1),T} + \rho^{-1}\alpha_Z^{(i)}\|_F^2
\tag{4}
$$

for some penalty parameter $\rho$. Lastly, $\alpha_Z$ is updated via

$$
\alpha_Z^{(i+1)} \leftarrow \alpha_Z^{(i)} + \rho(Z^{(i+1)} - [W^{(i+1)}, C](Q^{(i+1)})^T)
\tag{5}
$$

Equations 2 and 3 can be further split into row-wise constrained Lasso problems and there is a closed form solution for equation 4. The optimization details are further discussed in Appendix 6.1. Given the flexibility of ADMM, a similar procedure can also be used with other regularizations.

**Convergence of the optimization procedure**   The convergence hinges on the careful selection of the penalty parameter $\rho$. Informally, imposing the constraint $\rho \geq \sqrt{2}$ on the penalty parameter $\rho$ guarantees monotonicity of the optimization procedure, and that it will converge to a *local* minimum. Integrating this constraint with the adaptive selection of $\rho$ (Xu et al., 2017), we obtain an efficient optimization procedure for ICQF. Formally, this can be stated as the following proposition.

*Proposition* 3.1 (Non-increasing property).  Assume $\rho \geq \sqrt{2}$, we have

$$
0 \leq \mathcal{L}_\rho(W^{(i+1)}, Q^{(i+1)}, Z^{(i+1)}, \alpha_Z^{(i+1)}) \leq \mathcal{L}_\rho(W^{(i)}, Q^{(i)}, Z^{(i)}, \alpha_Z^{(i)}) \quad \forall i.
\tag{6}
$$

and by the monotone convergence theorem, $(W^{(i)}, Q^{(i)})$ will converge to a critical point $(W, Q)$.

The main idea of the proof of 3.1 is to estimate the difference between the two consecutive Lagrangians in Equation 6 by expanding it into

$$
\begin{aligned}
\mathcal{L}_\rho(\mathbb{V}^{(i+1)}, \alpha_Z^{(i+1)}) - \mathcal{L}_\rho(\mathbb{V}^{(i)}, \alpha_Z^{(i)}) =& \mathcal{L}_\rho(\mathbb{V}^{(i+1)}, \alpha_Z^{(i+1)}) - \mathcal{L}_\rho(\mathbb{V}^{(i+1)}, \alpha_Z^{(i)}) \\
& + \mathcal{L}_\rho(\mathbb{V}^{(i+1)}, \alpha_Z^{(i)}) - \mathcal{L}_\rho(\mathbb{V}^{(i)}, \alpha_Z^{(i)})
\end{aligned}
\tag{7}
$$

where $\mathbb{V}^{(i)} := \left\{ W^{(i)}, Q^{(i)}, Z^{(i)} \right\}$. Given that the subproblems $2 - 4$ are minimized during each iteration, we can estimate upper bounds of these terms and obtain

$$
\begin{aligned}
\mathcal{L}_\rho(\mathbb{V}^{(i+1)}, \alpha_Z^{(i+1)}) - \mathcal{L}_\rho(\mathbb{V}^{(i)}, \alpha_Z^{(i)}) \leq& \left( \frac{1}{\rho} - \frac{\rho}{2} \right) \cdot \Big( \|[W^{(i+1)}, C](Q^{(i+1),T} - Q^{(i),T})\|_F^2 \\
& + \|[(W^{(i+1)} - W^{(i)}), C]Q^{(i),T}\|_F^2 + \|Z^{(i+1)} - Z^{(i)}\|_F^2 \Big).
\end{aligned}
\tag{8}
$$

If we set $\rho \geq \sqrt{2}$, the right hand side becomes negative and the Lagrangian decreases across iterations and converges to a critical point. The full proof of Proposition 3.1 is given in Appendix 6.2.

Furthermore, Bjorck et al. (2021) showed that, for non-negative matrix factorizations, if the dimensionality $k$ is the same as that $k^*$ of a ground truth solution $(W^*, Q^*)$, the error $\|M - WQ^T\|_F^2$ is

star-convex towards $(W^*, Q^*)$, and the solution is close to a *global* minimum. However, if $k \neq k^*$, the relative error between $W^*$ and $W$ increases with $|\sqrt{k/k^*} - 1|$. Inaccurate estimation of $k^*$ thus affects both the interpretability of $(W, Q)$ and the convergence to global minima. With the bounded constraints imposed on $W$ and $Q$ in ICQF, Popoviciu's inequality establishes an upper bound for the variances $\sigma_W^2$ and $\sigma_Q^2$ of each column in $W$ and $Q$ respectively. To simplify the analysis, we assume equal variances among the columns (generally true). Then we have the following proposition:

*Proposition* 3.2. Let $(W^*, Q^*)$ be a ground-truth factorization of the given $\mathbf{M} = \mathbf{W}^*(\mathbf{Q}^*)^T$, with latent dimension $k^*$, where $\mathbf{W}^*$ and $\mathbf{Q}^*$ are matrix-valued random variables with entries sampled from bounded distributions. Suppose $(\mathbf{W}, \mathbf{Q})$ is another factorization with dimension $k \neq k^*$, then

$$\mathbb{E}\left[\|\mathbf{W}^* - \mathbf{W}\|_F^2\right] \geq \left(\sqrt{k/k^*} - 1\right)^2 \mathbb{E}\left[\|\mathbf{W}^*\|_F^2\right] \tag{9}$$

with high probability. The full proof of Proposition 3.2 is provided in Appendix 6.3. The two propositions, combined, show that our factorization can capture the true latent structure of the data, under the right conditions. The first is a linear combination of factors being a good approximation, which is the case for questionnaires. The second is having a robust estimator of $k$, discussed next.

**Choice of number of factors**     For each $\beta$, we choose the number of factors $k$ using blockwise-cross-validation (BCV). Given a matrix $M$, for each $k$, we shuffle the rows and columns of $M$ and subdivide it into $b_r \times b_c$ blocks. These blocks are split into 10 folds and we repeatedly omit blocks in a fold, factorize the remainder, impute the omitted blocks via matrix completion and compute the error[1] of that imputation. We choose $k$ with the lowest average error. This procedure can adapt to the distribution of confounds $C$ by stratified splitting. We compared this with other approaches for choosing $k$, for ICQF and other methods, over synthetic data, and report the results in Section 4.1.

# 4    Experiments and results

## 4.1    Experiments on synthetic questionnaire data

We examined the effectiveness of BCV and other algorithms on estimating the number of latent factors in a synthetic dataset, for ICQF against $\ell_1$-regularized NMF ($\ell_1$-NMF) (Cichocki & Phan, 2009) and factor analysis with promax rotation (FA-promax) (Hendrickson & White, 1964) as factors can be correlated. Both ICQF and $\ell_1$-NMF were initialized with NNDSVD (Boutsidis & Gallopoulos, 2008), and the sparsity ($\beta = 1\mathrm{e}{-1}$) and stopping criterion (relative iteration convergence tolerance $\epsilon < 1\mathrm{e}{-3}$) for fairness. The estimation method for FA was minimum residual.

We generated a synthetic questionnaire with $k^* = 10$ factors. We first created a $200 \times 10$ latent factor matrix $W$ (Figure 1 left). Each factor is present in isolation for 20 participants, and in tandem with another for 10 more, to synthesize correlation between factors. An entry of $W[i, j]$ is defined as

$$W[i, j] := D[i, j] \cdot a \cdot b, \quad a \sim U(0.5, 1), \ b \sim B(1, 0.9) \tag{10}$$

where $U(0.5, 1)$ is Uniform in $[0.5, 1]$ and $B(1, 0.9)$ is Bernoulli with probability $p = 0.9$.

Each factor had an associated loading vector – answer pattern – over 100 questions ($[0, 100]$ range). The resulting $100 \times 10$ loading matrix $Q$, shown in Figure 1 (center), is defined to be

$$Q[i, j] := c \cdot d, \quad c \sim U(0, 100), \ d \sim B(1, 0.3) \tag{11}$$

We then create a noiseless data matrix $M_{clean} := \min(0, \max(WQ^T, 100))$, and add noise by

$$M := \min\left(0, \max(M_{clean} + e \cdot f, 100)\right), \quad f \sim U(-100, 100) \tag{12}$$

where $e$ follows a discrete probability distribution with $P(e = 1) = \delta, P(e = 0) = 1 - \delta$. This yields a data matrix $M$, shown in Figure 1 (right) for $\delta = 0.3$ (the highest noise level).

---

[1]Appropriate weighting is multiplied to the error if number of blocks in the last fold is less than others.

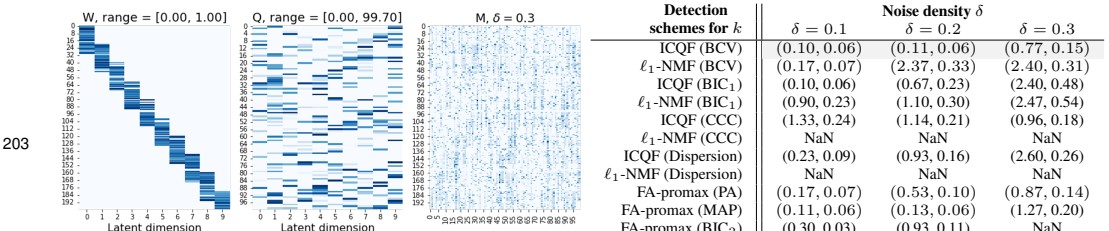

| Detection schemes for $k$ | Noise density $\delta$ | | |
|---|---|---|---|
| | $\delta = 0.1$ | $\delta = 0.2$ | $\delta = 0.3$ |
| ICQF (BCV) | (0.10, 0.06) | (0.11, 0.06) | (0.77, 0.15) |
| $\ell_1$-NMF (BCV) | (0.17, 0.07) | (2.37, 0.33) | (2.40, 0.31) |
| ICQF (BIC$_1$) | (0.10, 0.06) | (0.67, 0.23) | (2.40, 0.48) |
| $\ell_1$-NMF (BIC$_1$) | (0.90, 0.23) | (1.10, 0.30) | (2.47, 0.54) |
| ICQF (CCC) | (1.33, 0.24) | (1.14, 0.21) | (0.96, 0.18) |
| $\ell_1$-NMF (CCC) | NaN | NaN | NaN |
| ICQF (Dispersion) | (0.23, 0.09) | (0.93, 0.16) | (2.60, 0.26) |
| $\ell_1$-NMF (Dispersion) | NaN | NaN | NaN |
| FA-promax (PA) | (0.17, 0.07) | (0.53, 0.10) | (0.87, 0.14) |
| FA-promax (MAP) | (0.11, 0.06) | (0.13, 0.06) | (1.27, 0.20) |
| FA-promax (BIC$_2$) | (0.30, 0.03) | (0.93, 0.11) | NaN |

**Figure 1:** Synthetic $W$, $Q$ and $M$ with $\delta = 0.3$.  **Table 1:** Average error and standard error $(\bar{\epsilon}, s_E)$ of $k$.

Table 1 shows the mean error $\bar{\epsilon}$ and the standard error $s_E$ of the detected $k$ versus ground-truth $k^* = 10$, across 30 generated datasets. We tested five popular detection algorithms: BCV (Kanagal & Sindhwani, 2010), $BIC_1$ (Stoica & Selen, 2004)[2], CCC (Fogel et al., 2007) and Dispersion (Brunet et al., 2004). For ICQF and $\ell_1$-NMF, BCV is the best detection scheme at all noise levels; $BIC_2$ performs well for low noise only. For the three common FA schemes, Horn's PA (Horn, 1965) and MAP (Velicer, 1976) are superior to $BIC_2$ (Preacher et al., 2013), which aligns with empirical observations in Velicer et al. (2000); Watkins (2018); Goretzko et al. (2021). ICQF with BCV outperforms $\ell_1$-NMF and FA at all noise levels.

## 4.2 Experiments with the Child Behavior Checklist (*CBCL*) questionnaire

### 4.2.1 Data

The 2001 Child Behavior Checklist (*CBCL*) is a general-purpose questionnaire covering different domains of psychopathology designed to screen and refer patients to pediatric psychiatry clinics, for a variety of diagnoses (Heflinger et al., 2000; Biederman et al., 2005, 2020). The referral is based either on raw answers on the questionnaire or syndrome-specific subscales derived from them. The checklist includes 113 questions, grouped into 8 syndrome subscales: *Aggressive, Anxiety/Depressed, Attention, Rule Break, Social, Somatic, Thought, Withdrawn* problems. Answers are scored on a three-point Likert scale (0=absent, 1=occurs sometimes, 2=occurs often) and the time frame for the responses is the past 6 months. We use the parent-reported CBCL responses.

The primary experiments in this paper use CBCL questionnaires from two independent studies: the Healthy Brain Network (*HBN*) (Alexander et al., 2017) and the Adolescent Brain Cognitive Development[SM] (*ABCD*) study (https://abcdstudy.org). HBN is an ongoing project to create a biobank from New York City area care-seeking children and adolescents. ABCD is a longitudinal study, starting with youths aged 9-10, to obtain a socio-demographically representative sample over time. Both datasets provide diagnostic labels for mental health conditions, of which we selected the 11 most prevalent ones (Depression, General Anxiety, ADHD, Suspected ASD, Panic, Agoraphobia, Separation and Social Anxiety, BPD, Phobia, OCD, Eating Disorder, PTSD, Sleep problems). In HBN, we use CBCL from 1335 participants, 1,001 of whom have at least one diagnosis. In ABCD, we use CBCL from 11,681 participants, 7,359 of whom have at least one diagnosis.

### 4.2.2 Experimental setup

**Baseline methods**   Our first baseline method is $\ell_1$- regularized NMF ($\ell_1$-NMF) (Cichocki & Phan, 2009), as it also imposes non-negativity and sparsity constraints. As constructs (or questions) can be correlated, we rule out other NMF methods with orthogonality constraints. FA with promax rotation (FA-promax) (Hendrickson & White, 1964) using minimum residual as estimation method is included because it is the most commonly used technique for analyzing questionnaires and extracting latent constructs. It is also a baseline familiar to the clinical community designing questionnaires. Finally, syndrome subscales are included since they are often used for diagnostic prediction in screening. To estimate the number of factors $k$, we use BCV for $\ell_1$-NMF and ICQF, and Horn's parallel analysis for FA, the best approach for each method in the synthetic questionnaire experiments in Section 4.1.

**Dataset splits**   Within each dataset, we first split the participants into development and held-out sets with an 80/20 ratio. The assignment is done using stratified sampling, to keep the distribution of confounds and diagnostic labels similar across both sets. Training and validation sets are derived

---

[2]Here $BIC_1(k) := \log \left( \|M - WQ^T\|_F^2 \right) + k \frac{m+n}{mn} \log \left( \frac{mn}{m+n} \right)$, other versions yield similar results.

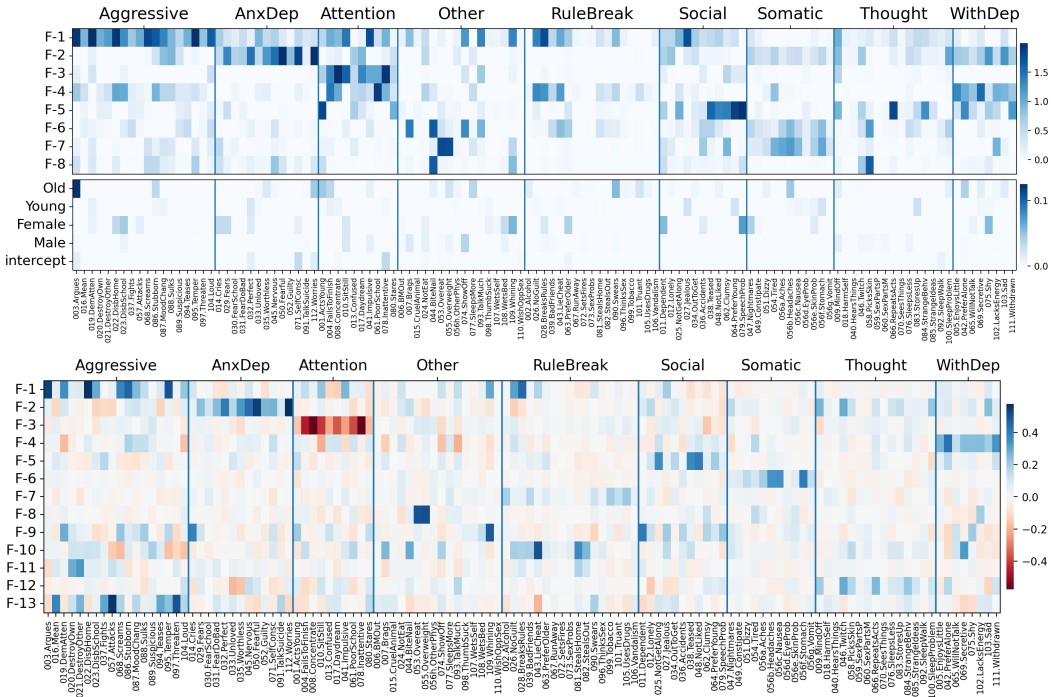

**Figure 2:** Heatmap of factor loadings $Q := [^R Q, ^C Q]$ from ICQF for factors proper, old/young and male/female confounds, and the implicit intercept (top) and loadings $Q$ from Factor Analysis with promax rotation (bottom). Abbreviated questions are listed at the bottom of each column. Questions are grouped by syndrome subscale; some factors are syndrome specific, while others bridge syndromes.

from the development set, as explained in each experiment. All the quantitative results are obtained on the held-out set. To increase the robustness of our analysis, and obtain measures of uncertainty, we use different seeds to resample 30 dataset splits, and carry out experiments on each split. The reported results are obtained by averaging the results on the held-out set across all 30 splits.

**Model training and inference** Let $W^{\text{set}}$ denote the participant factor matrix in ICQF or NMF, or the factor score in FA, with the superscript denoting the set. Similarly, let $Q$ denote the question loadings associated with a factor in each method. Model training will yield a $(W^{\text{train}}, Q)$ for participants in the training set. Inference with the model will produce $W^{\text{validate}}$ and $W^{\text{held-out}}$ in validation and held-out sets, using the trained $Q$ and confounds $C^{\text{validate}}, C^{\text{held-out}}$ (if applicable).

### 4.2.3 Experiment 1: qualitative comparison of ICQF with FA

We begin with a qualitative assessment of ICQF applied to the development set portion of the CBCL questionnaire from the HBN dataset. We estimated the latent dimensionality $k = 8$ using BCV to compute an error over left-out data, at each possible $k$. The regularization parameter $\beta = 0.5$ was set the same way. The top-panel of Figure 2 shows the heat map of the loading matrix $Q := [^R Q, ^C Q]$, composed of loadings $^R Q$ for the latent factors $W$, and the loadings $^C Q$ for the confounds $C$.

Given the absence of ground-truth factorizations for this questionnaire, the qualitative assessment hinges on the relation of question loadings to the syndrome subscales used in clinical practice. While there were factors that loaded primarily in questions from one subscale, as expected, we were encouraged by finding others that grouped questions from multiple subscales, in ways that were deemed sensible co-occurrences by our clinical collaborators. As a further, sanity check, we inspected the loadings of confound **Old** (increasing age) and observe that they covered issues such as *"Argues", "Act Young", "Swears" and "Alcohol"*. The loadings of $Q$ also reveal the relative importance among questions in each estimated factor; subscales deem all questions equally important.

For comparison, Figure 2 (bottom) shows the loadings $Q$ from Factor Analysis with promax rotation. By means of parallel analysis, we have identified a value of $k = 13$, which significantly exceeds the

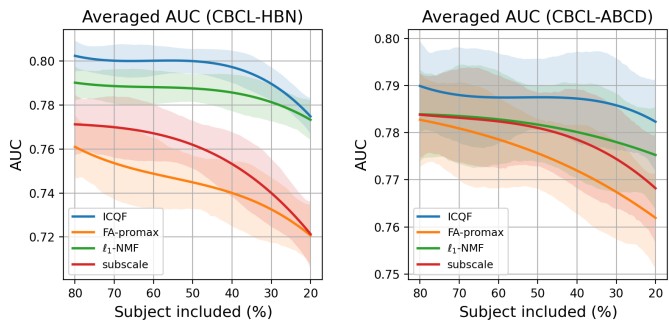

**Figure 3:** Trend and variability in average diagnostic prediction performance across 11 conditions, using decreasing dataset sizes, in CBCL questionnaires from HBN (left) and ABCD (right) independent datasets.

8 syndrome subscales that were initially established during the development of the checklist. The absence of sparsity and non-negativity control also results in a matrix that is more densely populated with both positive and negative elements, in an arbitrary range. This can present challenges when attempting to interpret the loadings in conjunction with the factor matrix $W$, also without constraints.

### 4.2.4 Experiment 2: preservation of diagnostic-related information

Our first quantitative metric to compare ICQF with baseline methods is the degree to which the low-dimensional factor representation of each participant (row of $W$) retains diagnostic information, across all 11 conditions we consider. Furthermore, this metric must be evaluated as a function of training sample size. As the sample size decreases, the regularization imposed by each method becomes more influential in determining the relationship between questions.

We evaluate this by creating training sets of different sizes from the development set (80, 40, 60, and 20 % of participants, with a fixed 20% as a validation set) and factorizing each of them with ICQF and the other methods. This yields a $W^{\text{train}}$, $Q^{\text{train}}$ for each combination of method and training set size, which is then used to infer factor scores $W_{\%}^{\text{held-out}}$ from the held-out set. The same held out-set is used for *every* method and dataset size being compared.

To estimate diagnostic prediction performance for each $W^{\text{train}}$, $Q^{\text{train}}$ factorization, we train a separate logistic regression model with $\ell_2$ regularization and balanced class weights from $W^{\text{train}}$ for each of the 11 diagnostic labels (i.e., 11 binary classification problems). The regularization strength is fine-tuned using $W^{\text{validate}}$, and prediction assessment is carried out on $W^{\text{held-out}}$ using the receiver operating characteristic (ROC) area under the curve (AUC) metric. The use of AUC is motivated from a clinical perspective, where clinicians often apply varying thresholds for detection depending on the aim of prediction, such as screening or intervention that incurs significant costs. We repeat this procedure in both CBCL-HBN and CBCL-ABCD data.

Figure 3 shows the trend and variability (95% confidence region) of the averaged AUCs of ICQF and the baseline methods using different dataset sizes (proportions of subjects), for HBN (left) and ABCD (right). In both HBN and ABCD, the ICQF outperforms other optimal baseline methods in maintaining high AUC scores across 11 conditions, and the difference in performance increases as the sample size decreases ($p \leq 0.01$, based on a one-side Wilcoxon signed rank test and adjusted using False Discovery Rate $\alpha = 0.01$), except for $\ell_1$-NMF at $20\%$ in CBCL-HBN). Moreover, the factorization solutions obtained with ICQF are more stable in terms of the number of dimensions $k$ ($k = 8 \rightarrow 6$ for ICQF, versus $8 \rightarrow 3$ for $\ell_1$-NMF and $13 \rightarrow 18$ for FA-promax in HBN; $k = 7 \rightarrow 7$ for ICQF, versus $5 \rightarrow 4$ for $\ell_1$-NMF and $20 \rightarrow 17$ for FA-promax in ABCD). This is particularly noteworthy in comparison to $\ell_1$-NMF, as it indicates the extra bounded constraints on $W$ and the approximation matrix $M_{approx}$ makes BCV detect $k$ more consistently.

### 4.2.5 Experiment 3: quality of the factor loadings

Our second quantitative metric to compare ICQF with baseline methods considers the change in quality of the factor loading matrix $Q$ as training sample size decreases, to examine the effect of regularization in constraining estimates. As before, we obtain a $W^{\text{train}}$, $Q^{\text{train}}$ for each combination

**Table 2: Top 2:** Quality of $Q$ factor loadings at various training set sizes, within dataset. The values are the mean and standard deviation of Pearson correlation coefficients between best-matched $Q$ factors from the full dataset, and from decreasing size subsets of it. Bolded where ICQF is significantly better. **Bottom:** Agreement in $Q$ factor loadings between models estimated in CBCL in two independent datasets, measured in the same way.

| Questionnaire | $n$-subjects | Factorization | | |
|---|---|---|---|---|
| | | ICQF | FA-promax | $\ell_1$-NMF |
| CBCL-HBN | 1854 (80%) | **0.89** (0.07) | 0.51 (0.41) | 0.76 (0.18) |
| | 1388 (60%) | **0.94** (0.03) | 0.62 (0.34) | 0.75 (0.19) |
| | 924 (40%) | **0.92** (0.05) | 0.62 (0.33) | 0.75 (0.19) |
| | 462 (20%) | 0.85 (0.12) | 0.54 (0.36) | 0.76 (0.20) |
| CBCL-ABCD | 7474 (80%) | **0.84** (0.13) | 0.43 (0.27) | 0.63 (0.28) |
| | 5604 (60%) | **0.84** (0.13) | 0.32 (0.30) | 0.63 (0.28) |
| | 3736 (40%) | **0.77** (0.20) | 0.42 (0.24) | 0.63 (0.28) |
| | 1868 (20%) | **0.69** (0.25) | 0.35 (0.26) | 0.62 (0.29) |
| CBCL-HBN $\leftrightarrow$ CBCL-ABCD | full $\leftrightarrow$ full | **0.75 (0.07)** | 0.71 (0.03) | 0.68 (0.08) |

of method and training set size. We then compare the loading matrix each size ($Q_\%$) with the one obtained on the full development dataset ($Q_{full}$). We do this by greedily matching each row from $Q_{full}$ with a row from $Q_\%$ by their Pearson correlation, and then computing the average correlation across pairs as the score. Given that a factorization learned on a smaller dataset may have fewer factors, we do this over the first $\min(k_{full}, k_\%)$ rows only. The first two rows of Table 2 reports this score for ICQF and the two baseline factorization methods, at each dataset size, on both CBCL-HBN and CBCL-ABCD datasets. ICQF outperforms the other methods at every dataset size ($p \leq 0.01$, based on a one-side Wilcoxon signed rank test and adjusted using False Discovery Rate $\alpha = 0.01$), except for $\ell_1$-NMF at 20% in CBCL-HBN.

Our third quantitative metric is the replicability of factor loadings across independent studies (and populations). This is an important criterion for clinical research purposes, as it means that the relations between questions identified by the factorization are general. We measure this by computing $W, Q$ for the full development sets of HBN and ABCD, for ICQF and the two baseline factorization methods. For each method, we greedily match factors loadings for the HBN and ABCD factorizations, and compute the average Pearson correlation across factor pairs, reported on the third row of Table 2. We conduct similar statistical testing and observe that ICQF outperforms the other methods ($p \leq 0.05$).

## 5   Discussion

In this paper, we introduced ICQF, a non-negative matrix factorization method designed for question-naire data. Our method incorporates characteristics that enhance the interpretability of the resulting factorization, as conveyed by psychiatry collaborators. We showed that their qualitative desiderata can be turned into formal constraints in the factorization problem, together with direct modelling of confounding variables, which other methods do not allow. The method is user friendly, by supporting automated estimation of the number of factors, minimizing the number of hyper-parameters, and transparently handling missing entries instead of requiring separate imputation. The characteristics above mean that ICQF required an entire optimization procedure to be derived from scratch. We provided a theoretical formalization of the problem and the procedure, and demonstrated a pair of propositions that guarantee convergence of the procedure to a local minimum and, in certain conditions, a global minimum as well.

We evaluated ICQF against alternative methods for the same purpose ($\ell_1$-NMF, used in the machine learning literature, and factor analysis, used in the clinical literature), on a widely used clinical questionnaire, in participants from two completely independent datasets. We designed metrics capturing the desired properties, namely preservation of diagnostic information – as this questionnaire is used for screening – and stability of solutions, at a range of dataset sizes, or across independent datasets. We carried out experiments controlling these factors, and showed that ICQF outperforms the alternative methods across the board. We have also used ICQF with 20 other questionnaires in HBN – both general-purpose and disorder-specific – in experiments not reported in this paper. Overall, results suggest that the regularization imposed by ICQF matches the underlying characteristics of questionnaire data better than other methods, in addition to promoting interpretability.

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
