# OpenReview forum: "Interpretable factorization of clinical questionnaires to identify latent factors of psychopathology"
_NeurIPS.cc/2023/Conference — Submitted to NeurIPS 2023_

### Official Review · Reviewer_gpT4 · 2023-06-26

**Soundness:** 4 excellent
**Presentation:** 3 good
**Contribution:** 3 good
**Rating:** 6
**Confidence:** 3

**Summary:**

This paper proposes a new technique to extract latent factors from psychological questionnaires. The method improves on previous methods both in terms of its ability to handle more flexible inputs (i.e., missing values and confounding variables) and in the interpretability of its outputs (i.e., the scale of its loadings are in the range of the original questionnaire). The proposed method is formulated as an optimization problem and an algorithm is provided to solve the problem and also to automatically determine an appropriate number of latent factors. Many experiments are provided that answer diverse sets of questions using both synthetic and real-world datasets.

**Strengths:**

1. The paper effectively formalizes psychology models as a solvable optimization problem that possesses better characteristics than competing methods.
2. The experiments are varied and show the usefulness across many scenarios: both when the true answer is known (synthetic data) and when it isn't.
3. Extensive comparison to existing techniques commonly used by researchers in the field of psychology.
4. They provide a Python implementation of their algorithm.

**Weaknesses:**

1. I realize space is tight, but many of the figures are incredibly small. Could some of the figures be moved to an appendix?


**Questions:**

1. Could the intuition behind the ICQF regularization method be explained more? The abstract talks about regularization specifically made for questionnaire data but I didn't see this explained beyond the function $R$ being defined. Were other regularization methods evaluated? How was this one chosen?
2. How does the complexity of the optimization problem scale with respect to $n$ participants or $m$ questions?
2. The paper is high-quality, and addresses a real problem in research, but I'm not entirely convinced NeurIPS is the best venue. Could you provide a small one to two sentence argument for why you think NeurIPS is an appropriate venue?

**Limitations:**

The paper effectively addresses common limitations with this kind of work by including a diverse set of experimental problems. If I were to give any feedback I'd say it might be worth discussing the inherent challenge in latent variable discovery. This would not be a weakness of your work but simply a challenge of the problem in general that you effectively addressed in the structure and design of the paper.

---

> ### Author Rebuttal · Authors · 2023-08-10
>
> ### Weaknesses
>
> > many of the figures are incredibly small. Could some of the figures be moved to an appendix?
>
> We thank the reviewer for pointing this out, and apologize for the incredibly small figure ticks and text due to page limit. To address this, we:
>
> - removed both $x$- and $y$-axis ticks' labels in Figure 1, as this figure only shows the general sample distribution. The full version together with an example of reconstructed result is added in the supplementary section.
> - removed the $x$-axis ticks' labels in Figure 2 and included a full version (rotated and enlarged) in supplementary section.
> - increased the font size of Figure 3.
>
>
> ### Questions
>
> > Could the intuition behind the ICQF regularization method be explained more?
>
> Thank you for bringing up this question. We agree that providing a more explicit explanation of the intuition behind the multiple constraints in our model is essential. These constraints were carefully defined in collaboration with our clinical partners to formalize objective and subjective characteristics that would make the factorization interpretable to them. We have attempted to provide a more detailed description of our motivation in this regard in the common response, as multiple reviewers asked about this. We ask the reviewer to please refer to this, and hope that the response will be satisfactory.
>
> > How does the complexity of the optimization problem scale with respect to $m$ participants or $n$ questions?
>
> Theoretically, ADMM exhibits linear convergence but, in practice, it often performs even faster. Typically, most experiments reported require only 50 iterations to reach the desired error tolerance. Within each iteration, we utilize the FISTA algorithm, known for its quadratic convergence, to solve subproblems 1 and 2. Subproblem 3 has a closed-form solution.
>
> Given that ADMM doesn't require every subproblem to reach the exact global minimum at each step, we set an upper bound of $K=20$ on the maximum number of iterations in FISTA. As a result, the overall complexity can be expressed in big O notation as $O(mnK/\epsilon)$, where $\epsilon$ represents the pre-defined error tolerance. It's worth noting that the FISTA algorithm can be parallelized row-wise, which can further enhance computational efficiency. Alternatively, we can opt to replace the FISTA algorithm with coordinate descent, especially for machines with few cores, as it would be more efficient in such cases.
>
> > Could you provide a small one to two sentence argument for why you think NeurIPS is an appropriate venue?
>
> Certainly! The aim of our research group is to develop machine learning methods to enable better scientific practice in psychology, psychiatry, and neuroscience. The technical characteristics of our factorization were developed in response to specific requests of our clinical collaborators; at the same time, we believe they are complex enough -- and novel enough, in combination -- to warrant publication in a machine learning venue. This is why we believe this work fits squarely within the  "Machine learning for sciences (e.g. climate, health, life sciences, physics, social sciences)" topic in the call for papers.

---

> > ### Comment · Reviewer_gpT4 · 2023-08-16
> >
> > I remain positive about the paper. On its face, the work appears to be very valuable for the author's target domain (i.e., factor analysis of psychology questionnaires). I do admit, however, that I am not incredibly versed in the most recent literature for this domain. When I do require it, my own work uses traditional PCA and Factor Analysis to analyze questionnaires.

---

> > > ### Author Response · Authors · 2023-08-16
> > >
> > > We thank the reviewer for their kind words. To our knowledge, the method of choice by default is still factor analysis, with some form of rotation. We have, on occasion, encountered uses of nonnegative matrix factorization, but without any of the additional constraints.

---

### Official Review · Reviewer_kCjw · 2023-06-28

**Soundness:** 2 fair
**Presentation:** 2 fair
**Contribution:** 2 fair
**Rating:** 3
**Confidence:** 3

**Summary:**

This paper proposes a matrix factorization formulation in order to extract latent features from questionnaires for psychopathology.

The proposed formulation is very similar to the well-known Non-Negative Matrix Factorization (NMF), with an l1-penalty on the dictionary and activation matrices. The biggest major different is they include some fixed dictionary vectors in the dictionary matrix. Due to some domain-related constraints they use the Alternating Direction Method of Multipliers (ADMM) method in order to impose the constraints.

The experimental results claim that the resulting factorization is interpretable. They also claim that the resulting factorization better preserves the clinical classification when compared to other factorization schemes such as l1-NMF and Factor Analysis. Finally the authors also claim that the proposed method better preserves correlation between the the activation matrix that is obtained from the full data and the activation matrix obtained from the subset of the data.


**Strengths:**

- The method seems appropriate in order to obtain latent factors from a questionnaire data.
- The method is lightweight.

**Weaknesses:**

- I am not really convinced that the proposed method is novel. It is simply an l1-constrained Non-Negative Matrix Factorization method. It is true that they add known variables inside the NMF dictionary, and impose additional value constraints, but in my opinion this does not seem like a novel method to me. Also, I do not think that the presented results present any novelty.
- The manuscript is hard to follow at times. For instance, I tried to understand how do the authors argue that the proposed factorization is interpretable, but the arguments put forth in section 4.2.3 in order to explain figure 2, remain difficult to understand for me. For instance the authors write that `While there were factors that loaded primarily in questions from one subscale, as expected, we were encouraged by finding others that grouped questions from multiple subscales, in ways that were deemed sensible co-occurrences by our clinical collaborators`. This sentence is hard to understand for me. I understand that maybe the authors' clinical collaborators might verbally confirm that these findings are sensible, but it would have really helped if the authors could do a user study to more quantitatively argue that their proposed method is superior compared to the other methods that they have compared against.
- In my opinion Figure 2 is critical in order to motivate the importance of your proposed method. It seems like there are some patterns with respect to different classes of psychopathologies, but I do not clearly understand what is the take away message here. The alternative method also seems to retain specific patterns for each class.
- You are comparing against two other linear factorization methods. I think it would have better to show that your method has advantages compared to deep neural network (e.g. an autoencoder with several layers).

**Questions:**

-Why did you limit the comparisons to linear factorization models? It is possible to construct a deep autoencoder and compare with the latent factors found that way. Given that this is the age of deep-neural networks, one in-avoidably thinks about using neural networks.
- I am not sure what is the motivation behind the experiment in section 4.2.5 ? More specifically I am not sure why reducing the number of subjects and measuring the correlation with the full data matrix is a good way overall to measure the factorization quality.
- I am not sure if I missed this in the manuscript, but did you try to understand what do the factors F-1, F-2 ... , F-8 correspond to?

**Limitations:**

The authors do not discuss negative societal impacts of their work, but I think this is okay, given that this work does not really pose dangerous implications.

---

> ### Author Rebuttal · Authors · 2023-08-10
>
> ### Weaknesses
> > The manuscript is hard to follow at times ... I understand that maybe the authors' clinical collaborators might verbally confirm that these findings are sensible, but it would have really helped if the authors could do a user study to more quantitatively argue that their proposed method is superior compared to the other methods that they have compared against.
>
> We apologize for the brevity of our writing, a result of page limitations, which might render the context less comprehensible, particularly for readers without a background in psychopathology.
>
> In the common response, we have provided a more detailed justification of why the constraints in our factorization promote "interpretability" as defined by our clinical collaborators. We ask the reviewer to please refer to that.
>
> The specific quoted sentence was meant to clarify that subscales, which are manual groupings of question items used in clinical practice, are used merely as a qualitative reference for evaluating the factorization. While subscales are undoubtedly useful, an active research question is whether they correspond to different underlying causes for the observed problems; alternatively, it's possible that the same cause leads to correlated answers across questions belonging to multiple subscales. Moreover, subscales cannot provide an indication of relative importance of different questions, which is also of interest to researchers.
>
> Factorizing a questionnaire matrix is a data-driven way of approaching this question, by seeing whether a given factor has loadings over questions in multiple subscales. As other reviewers had a similar question, we provide a detailed description of the ways in which the result of our method appears preferable to a clinical collaborator, versus that of factor analysis, in the common response section "Subjective evaluation". In the section "Motivation of the method", we also provide a more detailed explanation of how the constraints promote interpretability, as requested by our clinical collaborators.
>
> We agree with the reviewer that it would be ideal to have a comprehensive quantitative evaluation rooted in user studies involving clinical researchers. As it happens, this is something that is currently underway. This study encompasses not only the CBCL questionnaire in the HBN dataset, but also 21 selected questionnaires deemed pertinent to psychopathology. Our preliminary findings indicate that, across almost all questionnaires, the proposed methods yield more interpretable factor weights, which accurately convey the relative significance of different questions in terms of how informative they are for diagnosis.
>
> > It seems like there are some patterns with respect to different classes of psychopathologies, but I do not clearly understand what is the take away message here. The alternative method also seems to retain specific patterns for each class.
>
> Given our meticulous selection of baseline comparisons specific to our application's objectives, we expected to observe some patterns aligning with different classes of psychopathologies with either method. Questionnaires are designed assuming the existence of latent variables that combine linearly to produce observed answers.  The primary distinction of our method lies in making interpretation of the resulting solution easier, as described in "Motivation of the Method" in the common response. As above, we refer the reviewer to the "Subjective evaluation" section for an illustration of how a clinician might contrast the solutions provided by both methods.
>
>
> ### Questions
> > I am not sure why reducing the number of subjects and measuring the correlation with the full data matrix is a good way overall to measure the factorization quality.
>
> The primary objective of conducting this experiment was to simulate scenarios where the data size is small, a common occurrence in many psychology and psychiatry studies. Through this experiment, we sought to empirically demonstrate the robustness of our model to varying sample sizes. We observed that our model consistently maintains a certain level of latent factor interpretability, which we quantified by measuring its correlation with the latent factors discovered using the full data set. This consistent performance suggests that our model can be extended to scenarios where studies are conducted in different populations with similar sample distributions. Moreover, it also suggests that the regularization induced by our constraints matches the characteristics of the domain.
>
> > did you try to understand what do the factors F-1, F-2 ... , F-8 correspond to?
>
> Yes, our collaborators, who are experts in psychopathology, carefully examined each factor in detail, and found it easier to ascribe meaning relative to factors extracted by factor analyses. We refer the reviewer to section "Subjective evaluation" of the common response for more details. We apologize for not including the naming of factors assigned by domain experts due to page limitations. We have added the following table in Appendix in the modified manuscript.
>
> | Factor | Theme  |
> |---|---|
> | CBCL Factor-1 | irritability and oppositionality  |
> | CBCL Factor-2 | anxiety  |
> | CBCL Factor-3 | inattention and hyperactivity  |
> | CBCL Factor-4 | cognitive problems, disociality and callousness  |
> | CBCL Factor-5 | cognitive + fine motor problems  |
> | CBCL Factor-6 | body-focused repetitive behaviors  |
> | CBCL Factor-7 | somatic problems  |
> | CBCL Factor-8 | body-focused repetitive behaviors  |

---

### Official Review · Reviewer_LcU6 · 2023-07-06

**Soundness:** 3 good
**Presentation:** 3 good
**Contribution:** 2 fair
**Rating:** 3
**Confidence:** 4

**Summary:**

The paper proposes a non-negative matrix factorization with a customized regularization term to identify interpretable latent factors from psychopathological questionnaires. The input data is represented in a matrix and a non-negative matrix factorization algorithm is applied to the input matrix. The factor matrices are bounded to be between 0 and 1, providing some interpretation of the presence or absence of the corresponding factor.

**Strengths:**

The proposed method is overall sound. The optimization problem formulated and the regularization terms incorporated could well serve the desired purposes. The application of ADMM to solve the optimization problem is also reasonable. Good experimental results are shown using multiple datasets.

**Weaknesses:**

- The technical contribution is somewhat limited. Non-negative matrix factorization has been extensively studied for decades and widely applied to various applications, including questionary data analysis. ADMM is also a classic framework to solve matrix factorization problems with constraints. It seems to me that the optimization procedures described in Eq. (1-5) are standard for the ADMM algorithm and the convergence follows directly from the ADMM properties.
- The baselines used in the paper are very classic ones, $\ell_1$-NMF was developed in 2009 and FA-promax is developed in 1964. More recent methods for questionnaire data analysis should be compared.

**Questions:**

Please see the Weaknesses sections.

**Limitations:**

Limitations are not discussed in the manuscript.

---

> ### Author Rebuttal · Authors · 2023-08-10
>
> ### Weaknesses
>
> > The technical contribution is somewhat limited.
>
> > More recent methods for questionnaire data analysis should be compared.
>
> We ask the reviewer to please refer to the common response section for a detailed answer.

---

> > ### Comment · Reviewer_LcU6 · 2023-08-18
> > **Thank you for the response**
> >
> > Thank the authors for the response. I have carefully gone through them. Although I believe that the problem is well motivated, I respectfully disagree with the argument that there is no existing work that could incorporate the desired constraints such as probabilistic loading factors and handling missing data. In fact, there have been abundant papers in the past decades that incorporate those constraints/properties into NMF/PCA and apply them to different domains. Therefore, I will keep my rating unchanged.

---

> > > ### Author Response · Authors · 2023-08-18
> > >
> > > We thank the reviewer for taking the time to read through our responses. If we could impose further, we would appreciate pointers to the specific work that you mention, as we were not able to find any method with our combination of constraints, despite thorough searching  (as we hope the related work section would attest).

---

### Official Review · Reviewer_FBZh · 2023-07-07

**Soundness:** 3 good
**Presentation:** 3 good
**Contribution:** 2 fair
**Rating:** 4
**Confidence:** 4

**Summary:**

This paper presents an algorithm for factorizing matrix with multiple constraints desired in the study of psychiatric disorders using clinical questionnaires. These constrains include those that had been studied previously, such as sparseness in both the factor and loading matrices and non-negative values in both matrices. They also include two novel ones, that is the magnitude requirement on values in the matrix reconstructed from the factor and loading matrices and the value range requirement on the factor matrix (values in this matrix should be in [0, 1]). The authors also proposed to directly factor in confounding factors (e.g., age, gender) in the factorization. The algorithm was developed using the popular ADMM framework. Evaluation was done using both synthetic datasets and two practical datasets.

**Strengths:**

The results included in Table 2 is interesting, indicating the proposed method learns factors that are more stable than baseline methods with varying training set size. This is a desired behavior, implying it might learn the intrinsic patterns that are important.

**Weaknesses:**


Other than what’s mentioned in strength, the practical significance of this approach is limited given the existence of large amount of existing works in matrix factorization research. There is no clear statistical significance among results in Figure 3 to indicate obvious advantage of the proposed method over compared ones.

The advantage of including age and gender in the factorization is not clearly indicated. How with/without them affecting the learned factors is not clear.


**Questions:**

What is D in Eq. (10)? I was not able to find corresponding description.

---

> ### Author Rebuttal · Authors · 2023-08-10
>
> ### Weaknesses
> > the practical significance of this approach is limited given the existence of large amount of existing works in matrix factorization research.
>
> We ask the reviewer to please refer to the common response section for a detailed answer.
>
> > There is no clear statistical significance among results in Figure 3 to indicate obvious advantage of the proposed method over compared ones.
>
> We would like to point out that we observed statistically significant advantages of our method versus others as the sample size decreased, in both datasets, with the sole exception of $\ell_1$ NMF at 20% in CBCL-HBN.  However, our primary goal for this and other diagnostic prediction experiments was to show that having all the constraints that promote interpretability comes at *no* cost in terms of prediction performance i.e. our method preserves that information. This is something that our clinical collaborators deeply care about. These results suggest that the additional regularization proposed in our method matches domain characteristics.
>
> > The advantage of including age and gender in the factorization is not clearly indicated. How with/without them affecting the learned factors is not clear.
>
> The age and gender information are two of the most well-known potential confounder variables present in clinical data. Incorporating this information enables us to model answer patterns that are correlated with them. This prevents the creation of erroneous connections between question that are only associated due to these confounding variables. This distinction is crucial when drawing conclusions about the relationship between behavioral issues and diagnoses. This generalizes to other auxiliary variables, e.g. environmental exposures, life circumstances, etc.
>
> ### Questions
>
> > What is D in Eq. (10)? I was not able to find corresponding description.
>
> We apologize for not including the definition of $D$ in the manuscript. The matrix $D$ is a binary matrix in which columns are correlated with a step-like pattern, where each "step" is of length 20 and entries on the step have weight 1. Every consecutive pair of steps is overlapped by 10 units to synthesize correlation between latent factors. By multiplying random variables $a$ and $b$, we obtain the matrix $W$ as shown in the left panel of Figure 1.

---

> > ### Comment · Reviewer_FBZh · 2023-08-21
> >
> > Thanks to the authors for responding to my comments. Considering the lack of study on the impact of involving age and gender in the factorization and the limited novelty of the work as indicated by other reviewers, I keep my initial slightly negative rating.

---

> > > ### Author Response · Authors · 2023-08-21
> > >
> > > We thank the reviewer for taking the time to read through our responses. Regarding your comment, we would like to note that neither factor analysis nor sparse NMF explicitly involve age and gender, so we believe that would suffice to provide an indication that this affects performance.

---

### Author Rebuttal · Authors · 2023-08-10

We would like to thank all reviewers for taking the time to provide thoughtful feedback on our paper. We were pleased to see that, in general, reviewers agree that the paper is methodologically sound, clearly presented, and has an appropriate experimental evaluation.

Some issues have also been raised by multiple reviewers, specifically:
- motivation of the method
- technical contribution versus vanilla NMF methods
- subjective evaluation of interpretability
- the baseline methods compared against

To save reviewers time, we will cover these issues in a common response, prior to addressing individual reviewer comments and questions.

### Motivation of the method

Our method (ICQF) is motivated by psychological and psychiatric applications where questionnaires are the primary data type, and the goal is to make inferences about latent variables that correspond to constructs postulated by researchers. In this situation, interpretability and reproducibility across different datasets/populations are the two characteristics sought by them. While reproducibility is easy to quantify, interpretability is obviously subjective.

One of the contributions of the paper is to capture many of the characteristics that researchers told us would make a factorization interpretable to them as constraints in the method. Having factors in the [0,1] range means the factors can be interpreted as the degree to which the factor is present. Having the loadings for a factor be in the same scale as questions means that they can be interpreted as a pattern of answers, present in a participant to the degree the factor is present. Separately modelling confound variables means that their influence can be separated from that of the factors of interest. Constraining the reconstructed matrix not to exceed the possible range of the answers regularizes both factor and loading estimates, and contributes to their sparsity.

These characteristics may seem technically trivial, but *all* of them are missing from factor analysis (FA), the factorization method that has been the workhorse of psychological and psychiatric research for many decades. There, factors and loadings may be in an arbitrary range.  Interpretation of loadings requires taking into account the sign and range of the corresponding factor, as well as tradeoffs between positive and negative loadings. There is no allowance for confound or ancillary variables, and judging their influence requires dividing the sample by values or levels of the confound. Hence, our method should be viewed as a replacement for factor analysis, if one would start from scratch with researcher desiderata in mind. If there existed a non-negative matrix factorization method with all of these characteristics, we would have used it instead.

The other aspects of our method -- automated determination of dimensionality and sparsity, integrated handling of missing data -- are more practical in nature. They correspond, however, to steps that are challenging for researchers, and where practice is often ad-hoc. As we show in synthetic data experiments, our solutions outperform the approaches used in factor analysis at identifying the ground truth, and are completely integrated in the method rather than requiring separate steps and additional decisions by researchers.

Given these motivations, we believe this work fits squarely within the  "Machine learning for sciences (e.g. climate, health, life sciences, physics, social sciences)" topic in the call for papers.

### Technical novelty versus vanilla NMF methods

Our method was developed because we couldn't find any other non-negative factorization method that would satisfy all the constraints we wanted for factorization of a single questionnaire, as described above. We would not have embarked in developing a new method otherwise, especially given the need to prove convergence, examine performance with synthetic data, etc. This said, we understand reviewer concerns about technical novelty, and will try to address them here.

The components of the procedure, such as ADMM, are well understood; the novelty lies in their composition to achieve our specific goals.
The inclusion of bounded constraints for the factor matrix $W$ and the factor loading matrix $Q$ is essential for establishing convergence to a local minimum solution in Proposition 3.2.  Furthermore, we demonstrate that the combination of our method with the block cross-validation procedure can lead to a solution that is close to a global minimum. Both of these results are non-trivial, and necessary for the method to be practically applicable. Finally, the bounded constraints on $W$ enable a direct application to ICQF to concatenated factor matrices derived from many different questionnaires obtained in the same participants, as they will all be in the same scale. This is follow-up work we are doing with our collaborators to identify common dimensions of psychopathology manifesting across questionnaires. Empirically, we found that these extra constraints contribute to the stability of the factorization process and result in improved interpretability and robustness, particularly for small sample sizes.

[ Please proceed to "Author Rebuttal by Authors (Part II)" for further response. We apologize for splitting it into two posts.]

---

### Decision · Program_Chairs · 2023-09-21

**Decision:**

Reject

**Comment:**

This paper proposes a constrained NMF+regularization approach to learn interpretable latent factors for psychopathology. The analysis in the paper is quite comprehensive and the results are interesting in my opinion. The authors are motivated by interpretability and have not compared to deep learning baselines, while also focusing on linear models. As such this is well justified by the domain. However, there is a tension I see between this and methodologically comparing to a more comprehensive set of unsupervised learning baselines to test for the trade off of performance and interpretability. As such this limits the applicability of the paper.

Overall the reviewersaren't convinced of the novelty of the approach, and to some extent I agree. I have worked in constrained NMF myself in the healthcare context and the authors should extensively check on phenotyping literature some of which uses tensor and matrix factorizations (see [1] and [2]):

[1]Ho, J. C., Ghosh, J., & Sun, J. (2014, August). Marble: high-throughput phenotyping from electronic health records via sparse nonnegative tensor factorization. In Proceedings of the 20th ACM SIGKDD international conference on Knowledge discovery and data mining (pp. 115-124).

[2]Joshi, S., Gunasekar, S., Sontag, D., & Joydeep, G. (2016, December). Identifiable phenotyping using constrained non-negative matrix factorization. In Machine Learning for Healthcare Conference (pp. 17-41). PMLR.

Author rebuttals have extensively addressed all other clarifiying questions commendably. All things considered, reviews, accounting for author response, and my own read of the paper, it still is a challenging accept at NeurIPS.